# Janus Kinase Inhibitors Ameliorated Gastrointestinal Amyloidosis and Hypoalbuminemia in Persistent Dermatitis Mouse Model

**DOI:** 10.3390/ijms23010028

**Published:** 2021-12-21

**Authors:** Takehisa Nakanishi, Kento Mizutani, Shohei Iida, Yoshiaki Matsushima, Ai Umaoka, Makoto Kondo, Koji Habe, Keiichi Yamanaka

**Affiliations:** Department of Dermatology, Mie University Graduate School of Medicine, Edobashi Tsu 2-174, Mie 514-8507, Japan; takehisanakanishi@gmail.com (T.N.); k-mizutani@med.mie-u.ac.jp (K.M.); kmcasters@clin.medic.mie-u.ac.jp (S.I.); matsushima-y@clin.medic.mie-u.ac.jp (Y.M.); umaokaai@clin.medic.mie-u.ac.jp (A.U.); pjskt886@yahoo.ne.jp (M.K.); habe-k@clin.medic.mie-u.ac.jp (K.H.)

**Keywords:** inflammatory skin mouse model, dermatitis, cytokine, absorption, nutrition, emaciation, gastro-intestinal tract, amyloidosis, hypoalbuminemia, JAK inhibitor

## Abstract

Malnutrition is not only regarded as a complication of rheumatoid arthritis and inflammatory bowel disease but also that of inflammatory skin disease; however, the mechanisms and efficacy of its treatment have not been elucidated. Using a mouse model of dermatitis, we investigated the pathophysiology of malnutrition in inflammatory skin conditions and efficacy of its treatment. We employed spontaneous skin inflammation mice models overexpressing human caspase-1 in the epidermal keratinocytes. Body weight, nutrition level, and α1-antitrypsin fecal concentration were measured. The gastrointestinal tract was histologically and functionally investigated. Fluorescein isothiocyanate (FITC)-dextran was forcibly fed on an empty stomach, and plasma FITC-dextran was measured. The treatment efficacy of antibodies against tumor necrosis factor-α (TNF-α) and interleukin (IL)-α/β as well as Janus kinase (JAK) inhibitors was investigated. Compared with wild-type littermates, the inflammatory skin mice models showed a lowered body weight, reduction of serum albumin level, amyloid deposition in the stomach, small intestine, and large intestine, and increased α1-antitrypsin fecal concentration. However, the plasma FITC-dextran was unchanged between the dermatitis models and wild-type littermates. The over-produced serum amyloid A1 in the liver was detected in the plasma in the dermatitis model. Antibodies against TNF-α and IL-α/β showed partial effects on amyloid deposition; however, JAK inhibitors improved gastrointestinal amyloidosis with the improvement of skin symptoms. Chronic dermatitis is closely related to secondary amyloidosis in the gastrointestinal tract, resulting in hypoalbuminemia. Therefore, active control of skin inflammation is essential for preventing gastrointestinal complications.

## 1. Introduction

Patients with severe atopic dermatitis (AD) and erythroderma often have a tendency to lose weight and become undernourished. However, the detailed mechanism of emaciation associated with inflammatory skin conditions has not been elucidated; also, the therapeutic efficacy of antibodies against tumor necrosis factor-α (TNF-α) and interleukin (IL)-α/β as well as Janus kinase (JAK) inhibitors for emaciation is not clear. We considered that the cause may be protein leakage from the digestive tract. It is well known that chronic inflammatory diseases, such as rheumatoid arthritis (RA), inflammatory bowel disease (IBD), and systemic lupus erythematosus, cause secondary amyloid A (AA) amyloidosis [1]. Inflammatory cytokines, such as TNF-α and IL-6, produced during these chronic inflammatory diseases migrate to the liver leading to the production of serum amyloid A (SAA), causing multiple organ failure. On the other hand, the production of SAA from keratinocytes is enhanced by TNF-α, IL-17, and IL-22 [2]; subsequently, SAA may be deposited locally. SAA levels were positively correlated with the number of eosinophils in the peripheral blood of patients with AD [3]. 

AD is a chronic and intractable inflammatory skin disease. We have previously shown that the over-production of skin-derived inflammatory cytokines results in organ failure, such as cardiovascular sclerotic changes and cerebrovascular disorders, via persistent release of IL-1αβ from the inflammatory skin [4,5]. We conceived that inflammatory cytokines from persistent skin lesions also impact the gastrointestinal tract; however, the details have not been revealed. Here, using a mouse model of spontaneous dermatitis, we investigated the association between severe dermatitis, gastrointestinal amyloidosis, and hypoalbuminemia due to protein leakage. We also considered the therapeutic efficacy of antibodies against TNF-α and IL-α/β as well as JAK inhibitors for AA amyloidosis.

## 2. Results

### 2.1. Human Caspase-1 Gene with Keratin 14 Promoter (KCASP1Tg) Mouse Showed Emaciation

KCASP1Tg mice began to develop skin symptoms at around 8 weeks of age, starting on the face and subsequently spreading to the whole body [6]. The body weight (BW) and plasma albumin (Alb) from 6 to 16 weeks were determined. The BW of the KCASP1Tg mice was significantly lower than that of the wild-type (WT) mice (Figure 1a). Total protein (TP) levels were not significantly different between all the groups; however, plasma Alb was obviously decreased by phenotype, and hypoalbuminemia was recognized in the KCASP1Tg mice (Figure 1b–d). 

### 2.2. Histological Analysis Showed Amyloid Deposition in the KCASP1Tg Mice

At 16 weeks of age, the stomach, small intestine, and large intestine were collected from the KCASP1Tg and WT mice and stained with Congo red. Compared with the WT mice, the KCASP1Tg mice had more severe amyloid deposition in the gastric mucosa and intestinal epithelial cells. There was a significant increase in the Congo red stain-positive area in the KCASP1Tg mice (Figure 2).

### 2.3. Gastrointestinal Tract Barrier Function

Alpha1-antitrypsin concentration in feces was significantly more in the KCASP1Tg mice than in the WT littermates (Figure 3a). However, intestinal permeability was not considered to be enhanced by evaluating the plasma concentration of fluorescein isothiocyanate (FITC)-dextran in the KCASP1Tg mice (Figure 3b).

### 2.4. SAA Levels in Tissue and Plasma

The mRNA purified from various organs in the KCASP1Tg and WT mice revealed an increase in the SAA1 and SAA2 in the liver of the KCASP1Tg mice compared with that of the WT mice; furthermore, an increased level of SAA3 was detected in the skin of the KCASP1Tg mice (Figure 4a,b). The plasma levels of SAA1 were higher in the KCASP1Tg mice than in the WT mice; however, the plasma levels of SAA3 were similar in the two groups (Figure 4c,d).

### 2.5. The Effect of Anti-TNF-α and IL-1α/β in the KCASP1Tg Mice

KCASP1Tg mice were treated with neutralizing antibodies against TNF-α or IL-1α/β to prevent gastrointestinal amyloidosis, and amyloidosis was partially ameliorated (Figure 5).

### 2.6. The Effect of Administration of Janus Kinase (JAK) Inhibitors in KCASP1Tg Mice

KCASP1Tg mice were treated with baricitinib or cerdulatinib from 8 to 16 weeks of age (baricitinib and cerdulatinib, respectively), and the skin lesions were suppressed with these JAK inhibitors, especially with cerdulatinib (Figure 6a). Plasma Alb level was restored in the baricitinib- and cerdulatinib-treated KCASP1Tg mice (Figure 6b). At 16 weeks of age, the stomach, small intestine, and large intestine were collected and stained with Congo red. The severe amyloid deposit was more reduced in the gastric mucosa and intestinal epithelial cells of the baricitinib- and cerdulatinib-treated KCASP1Tg mice than in those of the non-treated KCASP1Tg mice (Figure 6c).

## 3. Discussion

The skin is one of the largest immune organs and functions as an alarmin, releasing various inflammatory and pro-inflammatory cytokines in response to external as well as intrinsic stimuli, in addition to its barrier function against the external pathogens. It has been suggested that persistent dermatitis may be complicated by atherosclerosis due to the release of cytokines from inflammatory sites in the skin [4]. Atherosclerosis was detected not only in the abdominal aorta but also in the peripheral basilar arteries, and these arteries were partially ameliorated by the administration of antibody preparations [5]. Cytokines produced locally in the skin can reach the adipose tissue of the abdomen, leading to the burning of fat cells and the release of adipocytokines, which contribute to the systemic inflammatory cascade [7]. Statistics have shown that psoriasis vulgaris and AD, which are examples of inflammatory skin diseases, have a high complication rate of coronary artery disease and cerebrovascular disease and are often fatal [8,9,10,11]. In inflammatory skin conditions, it has also been reported that osteoporosis may be a complication due to a decrease in the vascular network of the bone, an increase in osteoclasts, and a decrease in osteoblasts [12]. Male infertility is related to sperm hypoplasia caused by an increase in inflammatory cytokines from skin lesions [13]. In any case, we are still waiting for the development of drugs that can stop the cascade of inflammatory skin march that begins with dermatitis [14].

In the current study, we demonstrated that skin inflammation in mice models resulted in gastrointestinal amyloidosis and hypoalbuminemia. There was weight loss and decreased plasma Alb level, and one of the causes was considered to be protein leakage due to intestinal amyloidosis. Hypoalbuminemia is regulated by the balance between absorption disability and protein loss. Hypoalbuminemia occurs in various diseases, including liver cirrhosis and malnutrition, and it results in increased mortality. Chronic inflammatory diseases, such as rheumatoid arthritis, inflammatory bowel disease, and inflammatory autoimmune disease, also induce secondary hypoalbuminemia. The excessive cytokines produced in persistent dermatitis may be similar to that during systemic inflammation in rheumatoid arthritis. Hypoalbuminemia was detected even in the early phase in the KCASP1Tg mice. KCASP1Tg mice showed an obvious deposition of amyloid in the gastric mucosa and intestinal epithelial cells at three months (data not shown). 

Increased fecal levels of alpha 1-antitrypsin is an established marker of protein leakage syndrome caused by the abnormalities of lymphatic vessels, increased capillary permeability, or gut epithelium damage. Alpha 1-antitrypsin administration can be beneficial in the context of chemically induced colitis in mice [15]. The lymphatic vessels abnormality and gut epithelial inflammation were not obvious form the H&E section, and then increased capillary permeability may be the speculated causative reason. Alternatively, the hypoalbuminemia may result from liver-damaging inflammation, as suggested previously [4]. However, there was no significant difference between KCASP1Tg and wild-type mice in serum liver transaminase, cholinesterase, and total cholesterol levels; therefore, we can assume that liver function is largely intact in KCASP1Tg mice in the current study (data not shown). The fluorescently labeled dextran (FITC-dextran) model is well established for the evaluation of absorption/permeation pathways of substances through gastrointestinal epithelial cells: the short intercellular space pathway (paracelluar) and the transcellular pathway (transcelluar). High molecular weight FITC-dextran is absorbed/permeated by the short intercellular pathway. When epithelial cells are damaged, the absorption/permeation of dextran increases due to the loosening of intercellular junctions, and the greater the damage, the higher the amount of dextran absorbed/permeated. The present study suggests that severe damage to gastrointestinal epithelial cells or disruption of the short intercellular pathway is unlikely, and hypoalbuminemia from dermatitis may be due to increased capillary permeability; of course, additional experiments are required.

SAA is a family of apolipoproteins that are constitutively produced in several organs. SAA is also an acute phase protein produced in response to or enhanced by inflammatory stimulation, including pro-inflammatory cytokines IL-1, IL-6, and TNF-α, especially in the liver [16,17]. Acute phase SAA has several functions, such as inducing the recruitment of immune cells to inflammation sites and the transportation of cholesterol to the liver. In the KCASP1Tg mice, although SAA3 was overproduced in the inflamed skin lesions, the SAA3 level in the plasma did not increase. However, SAA1 and SAA2 production was enhanced in the liver of the KCASP1Tg mice. This was probably due to the increased circulating inflammatory cytokines produced from the active dermatitis, which activated circulating inflammatory cells, and SAA1 increased in the plasma.

In the current model, we focused on the prevention of gastrointestinal amyloidosis using neutralizing antibodies against inflammatory cytokines TNF-α and IL-1α/β; however, amyloidosis was partially ameliorated in both groups. Several reasons arose, and one possibility was that the 10 μg/body of anti-mouse TNF-α and IL-1α/β antibody administered to the KCASP1Tg mice three times a week was small. Since dermatitis is severe, a large amount of antibodies against cytokines are required. Alternatively, other candidates may accelerate gastrointestinal amyloidosis. Then, we administered JAK inhibitors to the KCASP1Tg mice. Baricitinib is a selective JAK1 and JAK2 inhibitor, and cerdulatinib hydrochloride is an orally active, multi-targeted tyrosine kinase inhibitor, especially for JAK1, JAK2, JAK3, and TYK2. The skin eruption was suppressed in JAK-treated mice models, significant amyloidosis was ameliorated, and plasma Alb level was recovered in JAK inhibitor-treated KCASP1Tg mice. 

In addition to above tested drugs, there are other candidates for the treatment of amylodosis. The mitogen-activated protein kinase (MAPK) family, key regulators of pro-inflammatory cytokines biosynthesis at the transcriptional and translational levels, is activated by a wide range of cellular stresses as well as in response to inflammatory cytokines, playing a central role in various biological phenomena such as cell proliferation, differentiation, transformation, survival, and apoptosis [18]. Similarly, inhibitors of apoptosis proteins (IAPs), a family of proteins that function as intrinsic regulators of the caspase cascade and have a crucial role on inflammatory skin conditions, may be considered to be the therapeutic target in the pathological development of inflammatory diseases. Recently, the gut–brain–skin axis theory has been focused to explain the interrelationship of gut microbiota, emotional states, and skin inflammation [19]. In fact, in the current model, there was affinity between skin inflammation severity and certain gut bacteria leading to a vicious cycle: skin inflammation populates certain gut bacteria, which itself worsens the skin inflammation [20]. Gastrointestinal amylodosis may contribute to bacterial dominance. Possible application of nutraceuticals, antioxidants, and anti-inflammatory compounds acting on the molecular signaling pathway directly or indirectly may be explored in the future.

Chronic dermatitis is closely related to secondary amyloidosis in the gastrointestinal tract, resulting in hypoalbuminemia. Therefore, the active control of skin inflammation is essential for preventing gastrointestinal complications. 

## 4. Materials and Methods

### 4.1. Mouse Study

Six-to 24-week-old female transgenic mice with keratinocytes specific caspase-1 overexpressing KCASP1Tg [6] and C57BL/6N littermate (WT) mice were used. The experimental protocol was approved by the Mie University Board Committee for Animal Care and Use (#22-39-5-1). BW was measured every 2 weeks from 6 weeks of age, and the mice were sacrificed at 24 weeks of age.

### 4.2. Blood Sampling and Clinical Chemistry Parameters

All the mice were euthanized with CO_2_ or pentobarbital. Blood was sampled from the tail vein or by cardiac puncture, placed in a 1.5 mL tube containing heparin, and centrifuged (6000 rpm for 5 min) to separate the plasma. The collected plasma was stored at −80 °C until examination. From 6 to 24 weeks of age, the concentrations of TP and Alb were measured using an automated analyzer. We also measured plasma SAA1 and SAA3 concentrations using enzyme-linked immunosorbent assay (ELISA) (Abcam, Cambridge, UK) at 16 weeks of age.

### 4.3. Gastrointestinal Tract Sampling and Congo-Red Stain

At 16 weeks of age, the stomach was collected from the cardia portion of the gastric body, the small intestine was excised from 1 cm proximal to the ileocecum to the first portion of the duodenum, and the large intestine was collected 1 cm from the rectum to the proximal side. All the samples were stored in a 10% neutral buffered formalin solution. Congo-red staining was performed on the stomach, small intestine, and large intestine tissue samples of the 16- and 24-week-old mice, and the stained sections were measured using ImageJ (https://imagej.nih.gov/ij/index.html (accessed on 16 September 2021). The percentage of Congo-red stain-positive lesions was calculated.

### 4.4. Gastrointestinal Tract Barrier Function

To evaluate absorption disability, feces were collected from mice aged 12 to 16 weeks, proteins were extracted using RIPA buffer (Fujifilm Wako Pure Chemical Corporation, Osaka, Japan), and alpha1-antitrypsin concentration in feces was measured by ELISA (Elabscience, Houston, TX, USA). To check for gastrointestinal tract barrier insufficiency, FITC-dextran (Sigma-Aldrich, St. Louis, MO, USA) was forcibly fed on an empty stomach from 12 to 16 weeks of age, and FITC-dextran in plasma was measured using a fluorometer. The experiment was performed in duplicate, and the absorbance was measured using Multiskan Jx (Thermo Fisher Scientific, Worsham, MA, USA). Values were analyzed using Ascent Software for Multiskan Ascent (Thermo Fisher Scientific). 

### 4.5. Real-Time Polymerase Chain Reaction (Real-Time PCR) and ELISA for SAA

Total RNA was extracted from several organs of the mice at 16 weeks of age using Tri Reagent (Molecular Research Center, Cincinnati, OH, USA). RNA concentration was measured using a NanoDrop Lite spectrophotometer (Thermo Fisher Scientific), and 1 µg total RNA was converted to cDNA using a High-Capacity RNA-to-cDNA Kit (Applied Biosystems, Foster City, CA, USA). The Taqman Universal PCR Master Mix II with uracil-N-glycosylase (Applied Biosystems) was used to measure the mRNA expression of SAA1 (Mm00656927_g1), SAA2 (Mm04208126_mH), and SAA3 (Mm0441203_m1). Glyceraldehyde-3-phosphate dehydrogenase (GAPDH Mm99999915_g1) was used as an internal control. All probes were purchased from Applied Biosystems, and amplification was performed using a LightCycler 96 System (Roche Diagnostics, Indianapolis, IN, USA).

### 4.6. Administration of Anti-TNF-α and IL-1α/β in the KCASP1Tg Mice

From 6 to 16 weeks of age, female KCASP1Tg and WT littermate mice were treated with 10 µg/body of anti-mouse TNF-α (BioLegend, San Diego, CA, USA) or IL-1α/β monoclonal antibody (BioLegend) three times per week intraperitoneally (*n* = 4, respectively). Phosphate-buffered saline (PBS) was injected as a control (*n* = 5). All the mice were sacrificed at the age of 16 weeks, and the gastrointestinal tract was investigated. 

### 4.7. Oral Administration of JAK Inhibitors

Eight-week-old female KCASP1Tg and WT littermate mice were orally treated with baricitinib (OYC1, Oriental Yeast, Kyoto, Japan) or cerdulatinib (Astellas, Tokyo, Japan). The treatment schedule was as follows: 5 mg/kg baricitinib or 1 mg/kg cerdulatinib was administered daily (*n* = 4) [21]. All the mice were sacrificed at 16 weeks of age, and the gastrointestinal tract was investigated.

### 4.8. Statistical Analyses

The text continues here. Statistical analyses were performed using PRISM software version 8 (GraphPad, San Diego, CA, USA). Two-group comparisons were analyzed using the Mann–Whitney test, and three groups or more were analyzed using the Mann–Whitney test or ordinary one-way analysis of variance. Differences were considered significant at *p* < 0.05, * *p* < 0.05, ** *p* < 0.01, *** *p* < 0.001, and **** *p* < 0.0001.

## 5. Conclusions

Generally, these results suggest that persistent dermatitis caused gastrointestinal amyloidosis, thus affecting nutrition. Although JAK inhibitors improve amyloidosis, active control of dermatitis is recommended to avoid gastrointestinal amyloidosis.

## Figures and Tables

**Figure 1 ijms-23-00028-f001:**
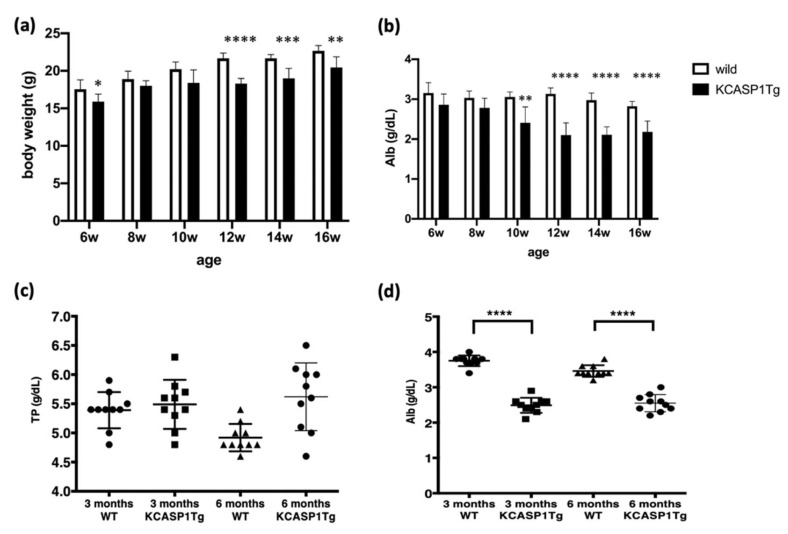
KCASP1Tg mice showed weight loss and hypoalbuminemia. (**a**) Body weight of the KCASP1Tg mice decreased significantly more than that of the WT littermates from 12 weeks of age. (**c**) TP was not significantly different between all the groups; however, Alb decreased from 10 weeks of age in the KCASP1Tg mice (**b**,**d**). WT; wild-type C57BL/6, KCASP1Tg; K14/caspase-1 transgenic mouse. Values shown as means ± SDs and statistically significant differences (* *p* < 0.05, ** *p* < 0.01, *** *p* < 0.001, **** *p* < 0.0001) are indicated.

**Figure 2 ijms-23-00028-f002:**
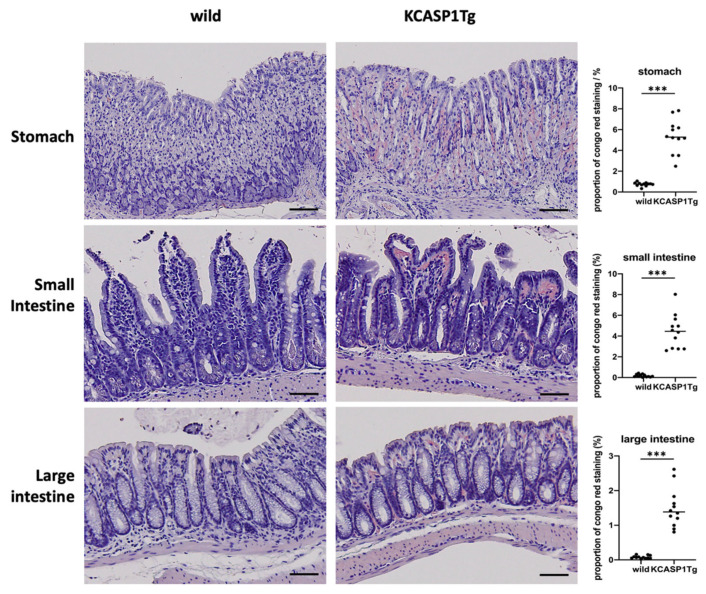
Histological analysis showed the amyloid deposition in KCASP1Tg mice. At 16 weeks of age, the stomach, small intestine, and large intestine were collected from the WT and KCASP1Tg mice and stained with Congo red. Compared with the WT mice, the KCASP1Tg mice had more severe amyloid deposition in the gastric mucosa and intestinal epithelial cells. Scale bar = 50 μm. There was a significant increase in the Congo red stain-positive area in the KCASP1Tg mice. All data were expressed as means ± SDs. *** *p* < 0.001 between KCASP1Tg and wild-type mice by Mann–Whitney test.

**Figure 3 ijms-23-00028-f003:**
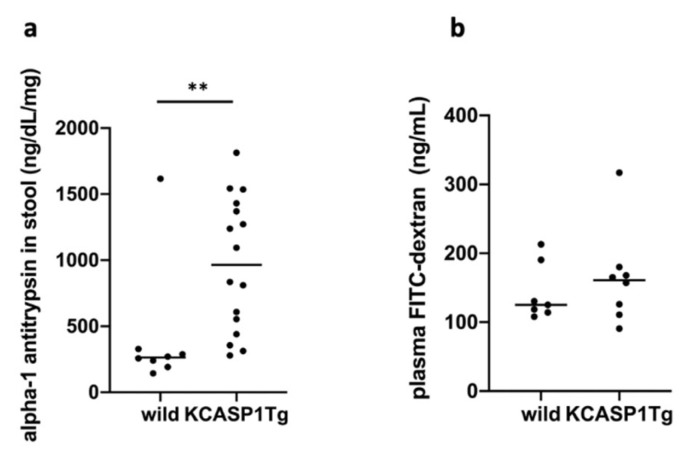
Gastrointestinal tract barrier function. (**a**) α1-antitrypsin in feces was significantly more in the KCASP1Tg mice than in the WT littermates (*n* = 6 per group). (**b**) However, intestinal permeability was not considered to be enhanced after evaluating the plasma concentration of FITC-dextran in the KCASP1Tg mice (*n* = 6 per group). All data were expressed as means ± SDs. ** *p* < 0.01 between KCASP1Tg and wild mice by Mann–Whitney test.

**Figure 4 ijms-23-00028-f004:**
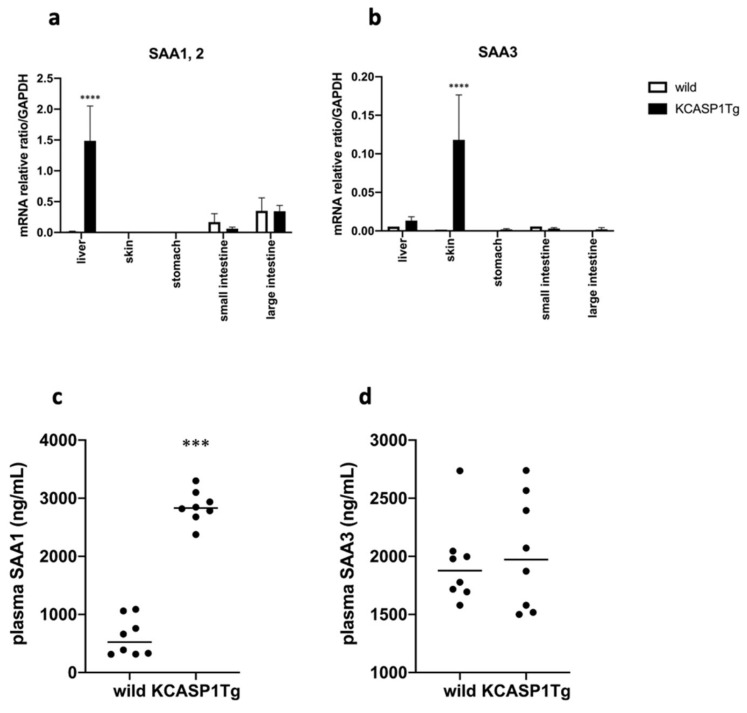
SAA levels in organs and plasma. (**a**) The mRNA purified from various organs in the KCASP1Tg and WT mice revealed that there was a higher production of SAA1 and SAA2 in the liver of the KCASP1Tg than in the WT mice. (**b**) An increased level of SAA3 was detected in the skin of the KCASP1Tg mice. (**c**,**d**) The plasma levels of SAA1 were higher in the KCASP1Tg mice than in the WT littermates; however, the plasma levels of SAA3 were similar between the two groups. All data are expressed as means ± SDs. *** *p* < 0.001, **** *p* < 0.0001 compared to wild mice by Mann–Whitney test.

**Figure 5 ijms-23-00028-f005:**
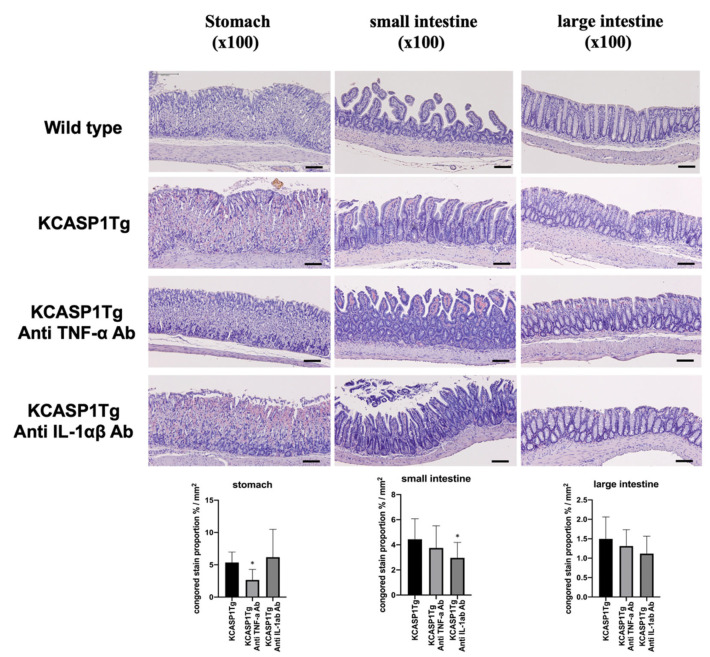
The effect of anti-TNF-α and IL-1α/β in the KCASP1Tg mice. KCASP1Tg mice were treated with neutralizing antibodies against TNF-α or IL-1α/β to prevent gastrointestinal amyloidosis, and the Congo red stain-positive area was partially ameliorated in the stomach and small intestine. Scale bar = 50 μm. * *p* < 0.05 compared to wild mice by Kruskal–Wallis test.

**Figure 6 ijms-23-00028-f006:**
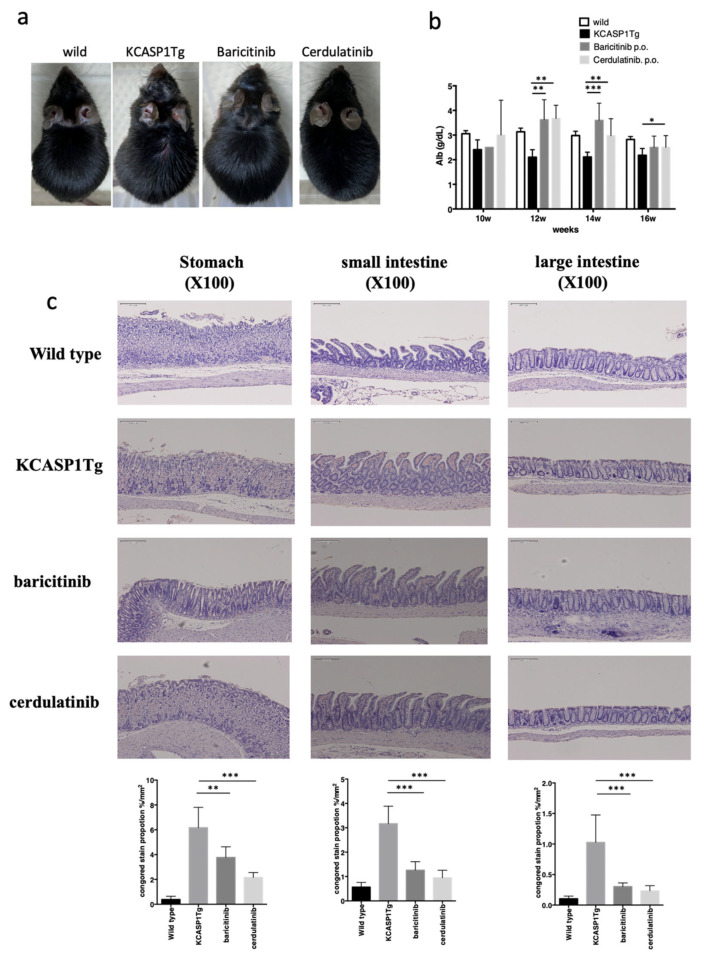
The effect of administration of JAK inhibitors in the KCASP1Tg mice. KCASP1Tg mice were treated with baricitinib or cerdulatinib (baricitinib and cerdulatinib, respectively). The clinical phenotype was ameliorated by both JAK inhibitors (**a**). Plasma Alb levels were measured from 10 to 16 weeks of age. Plasma Alb was recovered in baricitinib- and cerdulatinib-treated mice (**b**). At 16 weeks of age, the stomach, small intestine, and large intestine were collected and stained with Congo red. Amyloid deposition was reduced in the gastric mucosa and intestinal epithelial cells of the baricitinib- and cerdulatinib-treated KCASP1Tg mice compared with those of the non-treated KCASP1Tg mice (**c**). Scale bar = 100 μm. * *p* < 0.05, ** *p* < 0.01, *** *p* < 0.001 compared to KCASP1Tg mice by Kruskal–Wallis test.

## Data Availability

Not applicable.

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
