# Peer review of "Janus Kinase Inhibitors Ameliorated Gastrointestinal Amyloidosis and Hypoalbuminemia in Persistent Dermatitis Mouse Model"

_ijms, 2021, doi:10.3390/ijms23010028_

Round 1

Reviewer 1 Report

By utilizing the transgenic mouse in which inflammasome is constitutively active in the stratified squamous epithelia, Nakanishi et al. performed a descriptive study. 

Uncontrolled cutaneous inflammation could cause systemic inflammatory responses that lead to a range of disorders in the heart, the brain, and even in the bone. The authors focused on amyloid A (AA) proteins, which are produced locally and distantly.

The authors took a deep look at malnutrition caused by the skin-driven sustained inflammation.

Serum albumin levels and bodyweight of the KCASP1Tg mice were decreased substantially, which might be because of malabsorption from the gut epithelium in which AA deposits were in abundance.

The inflammatory phenotypes were reversed partially by anti-TNF/IL-1 therapy, and JAK/TYK inhibition appears to hold a promise for a better intervention for the systemic inflammation driven by the skin, a prime clinical example of which is the so-called "psoriatic march."

Although the data are presented nicely, the reviewer does not think that there are enough pieces of evidence that support the conclusion that malnutrition is attributable thoroughly to malabsorption from the gut epithelium, given the seemingly conflicting evidence in Fig. 3, based on the premises below. 

1. Increased fecal levels of alpha 1-antitrypsin (AAT) is an established marker of gut epithelium damage (including that caused by inflammatory assaults).

2. Gut inflammation causes leakiness and vice versa. 

3. The FITC-dextran model is a well-established method for analyzing gut leakiness.

4. AAT administration can be beneficial in the context of chemically induced colitis in mice (Collins et al. Alpha-1-antitrypsin therapy ameliorates acute colitis and chronic murine ileitis. Inflamm Bowel Dis. 2013;19: 1964-1973.)

Alternatively, the hypoalbuminemia may result from liver-damaging inflammation, as suggested previously (Yamanaka et al. Persistent release of IL-1s from skin is associated with systemic cardio-vascular disease, emaciation and systemic amyloidosis: the potential of anti-IL-1 therapy for systemic inflammatory diseases. PLoS One. 2014; 9:e104479.).

The discussion part seems redundant, somewhat repetitive, and can be more concise.

Author Response

Responses to the comments of Reviewer #1

#1   By utilizing the transgenic mouse in which inflammasome is constitutively active in the stratified squamous epithelia, Nakanishi et al. performed a descriptive study. Uncontrolled cutaneous inflammation could cause systemic inflammatory responses that lead to a range of disorders in the heart, the brain, and even in the bone. The authors focused on amyloid A (AA) proteins, which are produced locally and distantly. The authors took a deep look at malnutrition caused by the skin-driven sustained inflammation. Serum albumin levels and bodyweight of the KCASP1Tg mice were decreased substantially, which might be because of malabsorption from the gut epithelium in which AA deposits were in abundance. The inflammatory phenotypes were reversed partially by anti-TNF/IL-1 therapy, and JAK/TYK inhibition appears to hold a promise for a better intervention for the systemic inflammation driven by the skin, a prime clinical example of which is the so-called "psoriatic march."

Although the data are presented nicely, the reviewer does not think that there are enough pieces of evidence that support the conclusion that malnutrition is attributable thoroughly to malabsorption from the gut epithelium, given the seemingly conflicting evidence in Fig. 3, based on the premises below. 

  1. Increased fecal levels of alpha 1-antitrypsin (AAT) is an established marker of gut epithelium damage (including that caused by inflammatory assaults).
  2. Gut inflammation causes leakiness and vice versa. 
  3. The FITC-dextran model is a well-established method for analyzing gut leakiness.
  4. AAT administration can be beneficial in the context of chemically induced colitis in mice (Collins et al. Alpha-1-antitrypsin therapy ameliorates acute colitis and chronic murine ileitis. Inflamm Bowel Dis. 2013;19: 1964-1973.)

 Response: Thank you very much for reviewing my paper. The reviewer's comments were all on target, and we have modified our conclusions. We thank you for your useful and informative suggestions. We have been able to incorporate the four items you pointed out above into the discussion. We also cited the literature.

#2   Alternatively, the hypoalbuminemia may result from liver-damaging inflammation, as suggested previously (Yamanaka et al. Persistent release of IL-1s from skin is associated with systemic cardio-vascular disease, emaciation and systemic amyloidosis: the potential of anti-IL-1 therapy for systemic inflammatory diseases. PLoS One. 2014; 9:e104479.).

 Response: As the reviewer suggested, our previous report showed that in 6-month-old transgenic mice in which inflammasome is constitutively active in the stratified squamous epithelia, the liver architecture was disrupted by amyloid deposition, resulting in functional decline. However, in our 4-month-old mice, the clinical symptoms gradually decreased, as is often the case in transgenic mice, the liver damage was mild, and the transaminase level was unchanged from that of the wild type, so we speculated that the liver function is preserved in these mice. This is also incorporated in the discussion. We appreciated for your suggestion.

#3   The discussion part seems redundant, somewhat repetitive, and can be more concise.

Response: We have modified the discussion part. Thank you for the great suggestion.

Reviewer 2 Report

In the present paper, Nakanishi and coworkers aimed to delineate experimentally the pathophysiology of malnutrition in inflammatory skin conditions and efficacy of its treatment. Then, they concluded that, generally, persistent dermatitis caused gastrointestinal amyloidosis, thus affecting nutrition. Moreover, the authors suggest that, although JAK inhibitors improve amyloidosis, active control of dermatitis is recommended to avoid gastrointestinal amyloidosis.

Overall, I think that the paper could be of interest for readers of "International Journal of Molecular Sciences" and researchers, in general. I would like to make some suggestions on how to make the paper stronger.

1) MAPK (es. p38, p-ERK) signaling could represent a potential regulation pathway in the pathophysiology of malnutrition in inflammatory skin conditions. This feature could be careful considered in the paragraph of discussion.

2) Inhibitors of apoptosis (IAPs) are a family of proteins that function as intrinsic regulators of the caspase cascade and could have a crucial role on inflammatory skin conditions. Please make an appropriate comment about it.

3) Studies have shown that the gut microbiota may play a key role in the immunopathogenesis of inflammatory skin conditions and that essential pathways implicated are also regulated by the microbiota-gut-brain-skin axis. Please discuss this aspect and eventually take this factor into account in analysis of data.

4) In light of the results here obtained, please to discuss on the possible application of nutraceutics and/or antioxidants/anti-inflammatory compounds that, acting on molecular signaling pathway directly/indirectly explored in the present paper, could provide a possible strategy to prevent and counteract inflammatory cascade in patients affected by persistent dermatitis and gastrointestinal amyloidosis.

Author Response

Responses to the comments of Reviewer #2

In the present paper, Nakanishi and coworkers aimed to delineate experimentally the pathophysiology of malnutrition in inflammatory skin conditions and efficacy of its treatment. Then, they concluded that, generally, persistent dermatitis caused gastrointestinal amyloidosis, thus affecting nutrition. Moreover, the authors suggest that, although JAK inhibitors improve amyloidosis, active control of dermatitis is recommended to avoid gastrointestinal amyloidosis.

Overall, I think that the paper could be of interest for readers of "International Journal of Molecular Sciences" and researchers, in general. I would like to make some suggestions on how to make the paper stronger.

#1  MAPK (es. p38, p-ERK) signaling could represent a potential regulation pathway in the pathophysiology of malnutrition in inflammatory skin conditions. This feature could be careful considered in the paragraph of discussion.

Response: Thank you very much for reviewing our paper. We thank you for your useful and informative suggestions. This point has been included in the discussion part.

#2  Inhibitors of apoptosis (IAPs) are a family of proteins that function as intrinsic regulators of the caspase cascade and could have a crucial role on inflammatory skin conditions. Please make an appropriate comment about it.

Response: Above information has also been supplemented into the discussion.

#3  Studies have shown that the gut microbiota may play a key role in the immunopathogenesis of inflammatory skin conditions and that essential pathways implicated are also regulated by the microbiota-gut-brain-skin axis. Please discuss this aspect and eventually take this factor into account in analysis of data.

Response: We have been able to incorporate the suggestion you pointed out into the discussion. We appreciated for your comments.

#4  In light of the results here obtained, please to discuss on the possible application of nutraceutics and/or antioxidants/anti-inflammatory compounds that, acting on molecular signaling pathway directly/indirectly explored in the present paper, could provide a possible strategy to prevent and counteract inflammatory cascade in patients affected by persistent dermatitis and gastrointestinal amyloidosis.

Response: The suggestion you have pointed out was also included into the discussion. We appreciated for the great suggestion.

Round 2

Reviewer 1 Report

The authors addressed the concerns raised by the reviewer. However, the reviewer raises some concerns regarding the authors’ statement and response.

L178: Can the author provide any evidence that suggests that increased capillary permeability is responsible for protein loss? This appears entirely based on the assumption.

“in our 4-month-old mice, the clinical symptoms gradually decreased, as is often the case in transgenic mice, the liver damage was mild, and the transaminase level was unchanged from that of the wild type, so we speculated that the liver function is preserved in these mice. “

The liver is a vital producer of albumin, and transaminases can be a marker of liver injury, but not the function. The reviewer suggests that the simultaneous presence of hypergammaglobulinemia would be straightforward in Fg 1 if the sera are still accessible for the authors. 

Author Response

Responses to the comments of Reviewer #1

The authors addressed the concerns raised by the reviewer. However, the reviewer raises some concerns regarding the authors’ statement and response.

#1   L178: Can the author provide any evidence that suggests that increased capillary permeability is responsible for protein loss? This appears entirely based on the assumption.

  Response: Thank you very much for reviewing my paper. We toned down the discussion and conclusion, because as the reviewer suggested, the statement in this point is speculative. We thank you for your useful and informative suggestion.

#2   “in our 4-month-old mice, the clinical symptoms gradually decreased, as is often the case in transgenic mice, the liver damage was mild, and the transaminase level was unchanged from that of the wild type, so we speculated that the liver function is preserved in these mice. “ The liver is a vital producer of albumin, and transaminases can be a marker of liver injury, but not the function. The reviewer suggests that the simultaneous presence of hypergammaglobulinemia would be straightforward in Fg 1 if the sera are still accessible for the authors. 

 Response: Thank you for the great comments. We have additionally measured serum gammaglobuline level in transgenic mice and wild type mice, and gammaglobuline level was statistically increased in transgenic mice (supporting information). However, this elevation was enhanced by the increased serum IgG1, IgG2a, and IgG2b levels in transgenic mice, as reported previously (J Dermatol Sci. 2017 Oct;88(1):146-148). Instead, we also measured serum cholinesterase and total cholesterol levels in the current mice, and there was no statistical difference between transgenic mice and wild type mice (supporting information). Therefore, we can assume that liver function is largely intact in the current study. We appreciated for the great suggestion.

Reviewer 2 Report

The authors have satisfactorily responded to all my questions and made the necessary changes to the manuscript.

Author Response

Responses to the comments of Reviewer #2

The authors have satisfactorily responded to all my questions and made the necessary changes to the manuscript.

Response: Thank you very much for reviewing our paper. We thank you for your useful and informative suggestions.

Round 3

Reviewer 1 Report

The authors addressed the concerns raised by the reviewer enough, and the reviewer appreciates authors’s efforts and sincerlity.